# Functional and Phenotypic Characterization of Siglec-6 on Human Mast Cells

**DOI:** 10.3390/cells11071138

**Published:** 2022-03-28

**Authors:** Piper A. Robida, Clayton H. Rische, Netali Ben-Baruch Morgenstern, Rethavathi Janarthanam, Yun Cao, Rebecca A. Krier-Burris, Wouter Korver, Alan Xu, Thuy Luu, Julia Schanin, John Leung, Marc E. Rothenberg, Joshua B. Wechsler, Bradford A. Youngblood, Bruce S. Bochner, Jeremy A. O’Sullivan

**Affiliations:** 1Division of Allergy and Immunology, Department of Medicine, Feinberg School of Medicine, Northwestern University, Chicago, IL 60611, USA; parobida@nwosu.edu (P.A.R.); y-cao@northwestern.edu (Y.C.); rebecca.krier@northwestern.edu (R.A.K.-B.); j-wechsler@northwestern.edu (J.B.W.); bruce.bochner@northwestern.edu (B.S.B.); 2McCormick School of Engineering, Northwestern University, Evanston, IL 60208, USA; claytonrische2018@u.northwestern.edu; 3Division of Allergy and Immunology, Department of Pediatrics, Cincinnati Children’s Hospital Medical Center, University of Cincinnati College of Medicine, Cincinnati, OH 45229, USA; netali.benbaruch@cchmc.org (N.B.-B.M.); marc.rothenberg@cchmc.org (M.E.R.); 4Division of Gastroenterology, Hepatology, and Nutrition, Department of Pediatrics, Ann & Robert H. Lurie Children’s Hospital of Chicago, Chicago, IL 60611, USA; rjanarthanam@luriechildrens.org; 5Allakos, Inc., Redwood City, CA 94065, USA; wkorver@allakos.com (W.K.); alxu333@gmail.com (A.X.); tluu@allakos.com (T.L.); jschanin@allakos.com (J.S.); jleung@allakos.com (J.L.); byoungblood@allakos.com (B.A.Y.)

**Keywords:** mast cell, Siglec-6, FcεRI, degranulation, ITIM, endocytosis, signaling

## Abstract

Mast cells are tissue-resident cells that contribute to allergic diseases, among others, due to excessive or inappropriate cellular activation and degranulation. Therapeutic approaches to modulate mast cell activation are urgently needed. Siglec-6 is an immunoreceptor tyrosine-based inhibitory motif (ITIM)-bearing receptor selectively expressed by mast cells, making it a promising target for therapeutic intervention. However, the effects of its engagement on mast cells are poorly defined. Siglec-6 expression and endocytosis on primary human mast cells and mast cell lines were assessed by flow cytometry. *SIGLEC6* mRNA expression was examined by single-cell RNAseq in esophageal tissue biopsy samples. The ability of Siglec-6 engagement or co-engagement to prevent primary mast cell activation was determined based on assessments of mediator and cytokine secretion and degranulation markers. Siglec-6 was highly expressed by all mast cells examined, and the *SIGLEC6* transcript was restricted to mast cells in esophageal biopsy samples. Siglec-6 endocytosis occurred with delayed kinetics relative to the related receptor Siglec-8. Co-crosslinking of Siglec-6 with FcεRIα enhanced the inhibition of mast cell activation and diminished downstream ERK1/2 and p38 phosphorylation. The selective, stable expression and potent inhibitory capacity of Siglec-6 on human mast cells are favorable for its use as a therapeutic target in mast cell-driven diseases.

## 1. Introduction

Mast cells (MCs) are important cells in both innate and adaptive immunity [1,2]. They originate from a distinct lineage in the bone-marrow [3] and enter the circulation in an immature state, becoming fully mature once they have migrated to the tissues where they ultimately reside [4]. MCs can be found in virtually all organs of the body, including those that come into direct contact with external environments. They are best known for their role in allergic responses, where they can be activated upon allergen-crosslinking of IgE bound to its high-affinity receptors (FcεRI) that signal via immunoreceptor tyrosine-based activation motifs (ITAMs). The allergen/IgE/FcεRI pathway triggers the release of pre-stored and newly formed mediators that then elicit the signs and symptoms of allergic diseases ranging from sneezing to anaphylaxis [2]. In addition to FcεRI, MCs have numerous cell surface receptors whose engagement also results in activating responses including other ITAM-bearing receptors (e.g., FcγRIIA, CD300c, LILRA1), G-protein-coupled receptors (e.g., chemokine and complement receptors, MRGPRX2), MyD88-dependent receptors (e.g., IL1RL1 [ST2], Toll-like receptors), cytokine receptors (e.g., IL-3, stem cell factor) and others.

Besides activating receptors, additional transmembrane proteins serve to block or control MC activation. Indeed, the release of MC mediators can be prevented by a separate set of inhibitory receptors containing immunoreceptor tyrosine-based inhibitory motifs (ITIMs), such as FcγRIIB, CD300a, and several CD33-related sialic acid-binding immunoglobulin-like lectins (Siglecs), namely CD33, Siglec-7, and Siglec-8, via the dephosphorylation and inhibition of signaling molecules catalyzed by the recruited protein tyrosine phosphatases (PTPs) SHP-1/2 [5,6,7,8,9]. Co-aggregation of CD33, Siglec-7, or Siglec-8 with FcεRI, or engagement of Siglec-8 alone under certain circumstances markedly inhibits MC activation in vitro or MC-dependent anaphylaxis or inflammation in vivo in mouse models [10,11,12,13,14,15].

Siglec-6, originally identified on trophoblast cells of the placenta [16,17,18], is highly and consistently expressed on MCs from both mucosal and non-mucosal sites, as is Siglec-8 [7,14,19]. Siglec-6 appears to be selectively expressed by mast cells among non-malignant leukocytes [19] but is also expressed by some acute myeloid leukemia [20,21] and chronic lymphocytic leukemia cells [22] and cell lines, including U937 and THP-1 cells [16]. Like Siglec-8, Siglec-6 possesses a cytoplasmic ITIM and an immunoreceptor tyrosine-based switch motif (ITSM), each of which shares 83% sequence identity with its counterpart motif in Siglec-8, although overall sequence identity shared by Siglec-6 and Siglec-8 is slightly less than 50% [23,24]. Unlike Siglec-6 and Siglec-8, certain other receptors are expressed in a tissue-specific manner on MCs. For example, MRGPRX2 and CD171 (neural cell adhesion molecule L1) are only expressed on fat and skin MCs but not on lung or colon MCs [19].

While both Siglec-6 and Siglec-8 are intriguing therapeutic targets on MCs in all tissues [8], the functional outcomes of Siglec-6 engagement remain relatively unexplored in MCs compared to other Siglecs, with just a single publication showing very modest inhibition of degranulation in response to FcεRI receptor stimulation [25]. Ligands for Siglec-6 are similarly poorly characterized and appear to be distinct from those of all other Siglecs studied, with a few low-affinity ligands identified that lack specificity such as leptin, glycodelin-A, and the glycan Neu5Acα2–6GalNAcα (sialyl-Tn) [16,26,27]. Though the cytoplasmic signaling motifs in Siglec-6 have been found to be capable of being phosphorylated and recruiting SHP-2 [28], and the Siglec-6 interaction with glycodelin-A leads to ERK1/2 dephosphorylation [25] in placental trophoblast cell lines, the physiologic consequences of Siglec-6 engagement or co-engagement have not been studied in MCs. Therefore, the goal of the present study was to further characterize the expression and function of Siglec-6 on human MCs. The primary hypothesis being tested is that engagement of Siglec-6 will effectively counteract the function of receptors that activate MCs.

## 2. Materials and Methods

### 2.1. Skin Mast Cell Isolation and Culture

Primary human MCs were isolated from de-identified normal human skin specimens (obtained through the Cooperative Human Tissue Network) that were discarded from plastic surgery procedures. Briefly, skin specimens, with subcutaneous fat removed, were cut into small pieces and enzymatically digested in wash buffer (Hanks balanced salt solution (HBSS), 1% fetal calf serum (FCS), 10 mM 4-(2-hydroxyethyl)-1-piperazineethanesulfonic acid (HEPES), 0.035% sodium bicarbonate, 0.05% amphotericin B and 1% penicillin/streptomycin (all from ThermoFisher Scientific, Waltham, MA, USA)) supplemented with hyaluronidase (0.7 mg/mL; Sigma-Aldrich, St. Louis, MO, USA), type II collagenase (592.5 U/mL; Worthington, Lakewood, NJ, USA), DNase type I (0.15 mg/mL; Sigma-Aldrich), and 1 mM CaCl_2_ (Sigma-Aldrich). The specimens were digested on a shaker at 37 °C at 270 rpm for 1 h. The digested mixture was filtered through a wire mesh, and the remaining tissue was collected for further digestions. Collected cells were washed, spun down, filtered through a 40-μm filter, and washed again. The resulting pellet was resuspended in wash buffer and kept on ice. After three digestions, the collected pellets were combined and processed through a Percoll gradient. Cells were collected, washed, and plated at a concentration of 5 × 10^5^ cells/mL in serum-free X-Vivo medium (Lonza, Basel, Switzerland) supplemented with 100 ng/mL recombinant human SCF (Peprotech, Cranbury, NJ, USA). Cells were fed weekly and maintained at a concentration of 5×10^5^ cells/mL.

### 2.2. In Vitro Mast Cell Differentiation from CD34+ Cells

Human MC were generated from CD34+ hematopoietic stem cells isolated from whole blood as described [29]. After seven weeks in culture, cells were maintained in Iscove’s modified Dulbecco’s medium (IMDM) supplemented with 5% FCS, 55 μM β-mercaptoethanol, 100 ng/mL SCF, and 50 ng/mL IL-6.

### 2.3. Cell Lines and Culture

Several human MC lines were used in this study: HMC-1.2 cells were generously provided by P. Valent of the Medical University of Vienna (Vienna, Austria) [30], LUVA cells were generously provided by L. Borish of the University of Virginia School of Medicine (Charlottesville, VA, USA) [31], and the SCF-dependent MC line ROSA *KIT*^WT^ and the line transfected with *KIT*^D816V^, ROSA *KIT*^D816V^, were generously provided by M. Arock of Pitié-Salpêtrière Hospital (Paris, France) [32]. HMC-1.2 cells were cultured in IMDM supplemented with 25 mM HEPES, 3.024 g/L sodium bicarbonate, 10% FCS, 1% L-glutamine, 1% penicillin-streptomycin, and 0.1% β-mercaptoethanol (all from ThermoFisher Scientific). LUVA cells were cultured in StemPro-34 serum-free medium supplemented with the StemPro-34 nutrient supplement, 1% L-glutamine, and 1% penicillin/streptomycin. ROSA cells were cultured in IMDM supplemented with 1% sodium pyruvate, 2% MEM non-essential amino acids, 1% insulin-transferrin-selenium (ITS-G), 1% L-glutamine, 1% MEM vitamin solution, 1% penicillin-streptomycin, and 0.3% BSA (Sigma-Aldrich). The medium for the ROSA *KIT*^WT^ cells was supplemented with 80 ng/mL rhSCF, whereas the medium for the ROSA *KIT*^D816V^ cells was supplemented with 10% FCS. The acute myeloid leukemia cell lines THP-1 and U937 were generously provided by R. Pope of the Northwestern University Feinberg School of Medicine (Chicago, IL, USA) [33,34]. Both THP-1 and U937 cells were cultured in Roswell Park Memorial Institute 1640 medium supplemented with 10% FCS and 1% penicillin-streptomycin, with β-mercaptoethanol added at a final concentration of 0.05 mM to the THP-1 medium.

### 2.4. Single-Cell RNAseq

Single-cell suspensions were prepared from esophageal biopsies as previously described [35]. The single-cell suspensions were directly subjected to the 10× mass genomics chip (10× Genomics, Inc, Pleasanton, CA, USA), targeting 10,000 simultaneously captured live events for next generation sequencing. Each cell was uniquely barcoded and sequenced by the CCHMC DNA sequencing core on the Illumina HiSeq 2500, with total reads of ~320M (2 lanes/flow cell). All scRNA-seq data were processed using Cell Ranger version 3.0.2 and hg19 reference. The unique molecular identifier (UMI) count matrix was imported to Seurat for further analysis as previously described [36,37]. Cells were filtered based on unique feature counts > 4800 or <200, excluding cells with >20% mitochondrial counts. The UMI count matrix was normalized and scaled following the standard Seurat pipeline by adjusting for mitochondrial read counts and UMI counts. Principal component analysis was then performed over the list of variable genes, and the data were subjected to uniform manifold approximation and projection, and shared nearest neighbor modularity optimization-based clustering. The top expressed gene markers with high specificity were used to classify cells into seven known types: epithelium, fibroblast, endothelium, T cells, B cells, myeloid cells, and mast cells. Log normalization and centering were used to calculate relative expression. Detailed methods for the single-cell RNAseq work performed at Ann & Robert H. Lurie Children’s Hospital of Chicago are provided in the Appendix A for this article.

### 2.5. Siglec-6 Antibody Crosslinking or Co-Crosslinking

#### 2.5.1. Siglec-6 Engagement on HSMCs in Parallel with Stimulation

After at least 12 weeks of culture, HSMCs were pre-incubated with anti-Siglec-6 (clone 767329; R&D Systems, Minneapolis, MN, USA) or isotype control mAb (BioLegend, San Diego, CA, USA) at 5 μg/mL for 30 min, unless otherwise indicated, prior to stimulation with the indicated concentration of anti-FcεRIα (clone CRA-1; BioLegend), rhC5a (Peprotech), or compound 48/80 (Sigma-Aldrich). HSMC activation was then assessed by β-hexosaminidase release.

#### 2.5.2. Secondary Antibody Co-Crosslinking of FcεRIα and Siglec-6 on CD34+ Cell-Derived MCs

Cells were plated in 96-well round bottom tissue culture plates at 5 × 10^4^ per well and centrifuged for 2 min at 400 g prior to resuspending in anti-FcεRIα (clone CRA-1, BioLegend) at 150 ng/mL plus isotype antibody (clone MOPC21) or anti-Siglec-6 mAb (clone 767329, R&D Systems) at 5 μg/mL at 4 °C for 2 min. Cells were washed with PBS and then incubated in PBS with 5 μg/mL secondary antibody (goat anti-mouse IgG light chain specific, Jackson ImmunoResearch) for 2 min. After an additional PBS wash, cells were resuspended in warm complete medium and incubated for the indicated times at 37 °C.

#### 2.5.3. Streptavidin Complex Production and Co-Crosslinking of FcεRIα and Siglec-6 on HSMCs

Streptavidin complexes were created by mixing biotinylated anti-Siglec-6 (or isotype control) and anti-FcεRIα (or isotype control) mAbs at the indicated molar ratio (1:1, 3:1, or 9:1, anti-Siglec-6:anti-FcεRIα), then adding streptavidin at a 1:4 molar ratio and incubating at 4 °C overnight. Anti-Siglec-6 (clone 767329) was biotinylated using the Pierce Antibody Biotinylation Kit for IP (ThermoFisher Scientific). Biotin-conjugated anti-FcεRIα, mIgG2a isotype control mAb, and mIgG2b isotype control mAb were purchased from BioLegend. Complexes were produced such that the final treatment concentrations of anti-Siglec-6 and anti-FcεRIα in the 1:1 molar ratio complexes would equal 1.25 μg/mL, matching the concentrations of the free antibody treatments. HSMCs were incubated with streptavidin complexes or free antibody for 30 min at 37 °C before assessing cell activation and epitope blockade by flow cytometry.

### 2.6. Detection of Secreted Mast Cell Mediators, Cytokines, and Chemokines

Tryptase was quantified in cell-free supernatant using the Mast Cell Degranulation Assay Kit (EMD Millipore, Burlington, MA, USA). For cytokine and chemokine quantification, 25 μL of cell-free supernatant was analyzed using Meso Scale Discovery’s U-plex kit (customized for the quantification of the indicated cytokines and chemokines; Rockville, MD, USA) after 6 or 24 h incubation at 37 °C.

### 2.7. Flow Cytometry

To quantify the number of Siglec-6 receptors on the cell surface, Siglec-6 on human skin-derived MCs (HSMCs) was detected with anti-Siglec-6 mAb (R&D Systems). Similarly stained Quantum Simply Cellular Beads (Bangs Laboratories, Fishers, IN, USA) coated with anti-mouse IgG were used to generate a standard curve from which receptor numbers could be calculated from fluorescence intensity. MCs were stained with antibodies specific for Siglec-6 (clone 767329, R&D Systems), Siglec-8 (clone 2C4 [24]), CD117/KIT (YB5.B8; BD Biosciences, Franklin Lakes, NJ, USA), FcεRIα (CRA-1; BioLegend), CD63 (H5C6; BioLegend), and LAMP-1 (H4A3; ThermoFisher Scientific) in cold fluorescence-activated cell sorting (FACS) buffer (PBS/1% BSA). Isotype control mAbs (BioLegend) were used as negative staining controls, and Fc Block (BD Biosciences) was used to reduce nonspecific binding. Dead cells were excluded from analysis based on 7-AAD (BD Biosciences), DAPI, or LIVE/DEAD (ThermoFisher Scientific) staining. Phosphotyrosine levels in THP-1 cells were assessed using fluorophore-conjugated anti-phosphotyrosine mAb (clone PY2; BioLegend) after fixation and permeabilization of the cells with BD Cytofix/Cytoperm. Samples were run on a BD LSR II (Franklin Lakes, NJ, USA), Novocyte Quanteon (Agilent Technologies, Santa Clara, CA, USA), or a Beckman Coulter CytoFLEX (Brea, CA, USA) flow cytometer. Data were analyzed using FlowJo software v10 (TreeStar, Ashland, OR, USA). Receptor endocytosis was determined by delayed secondary staining as previously described [38]. Briefly, cell-surface Siglec-6 or Siglec-8 was bound with unlabeled mAb at 4 °C, and excess mAb was removed by two sequential wash steps. MCs were incubated at 37 °C for the indicated durations to permit receptor endocytosis prior to staining the remaining anti-Siglec-6/8 mAb with a fluorophore-conjugated secondary mAb and detecting secondary staining by flow cytometry. Initial levels of antibody binding were determined on cells not incubated at 37 °C. Cells stained with fluorophore-conjugated anti-Siglec-6/8 and similarly incubated were used to assess receptor shedding or lack thereof.

### 2.8. Intracellular Phospho-Flow Analysis

Cells were stimulated as indicated for 15 min at 37 °C. After treatment, cells were fixed by adding PFA to a final concentration of 3% and incubating at room temperature for 10 min. The cells were then washed in FACS buffer, permeabilized by resuspending in cold (−20 °C) methanol, added slowly while shaking, then incubated on ice for 30 min and stained with anti-phospho-ERK1/2 (Thr202/Tyr204, clone 197G2, Cell Signaling Technology, Danvers, MA, USA) or anti-phospho-p38 (Thr180/Tyr182, clone 28B10, Cell Signaling Technology). Cells were washed in FACS buffer and analyzed by flow cytometry.

### 2.9. Quantification of β-Hexosaminidase Release

MCs (at 12 weeks of culture) were activated for 30 min with different concentrations of anti-FcεRIα, rhC5a, or compound 48/80, following a pre-incubation (30 min or otherwise denoted) with anti-Siglec-6 or isotype control mAb. MC supernatants were collected and cell pellets were lysed with PBS/0.01% Triton-X. The amount of β-hexosaminidase in the supernatant was compared to the amount remaining in the cells by colorimetric assay.

### 2.10. Statistical Analysis

Data are presented as means ± standard deviations unless otherwise indicated. Statistical significance was determined by one-way or two-way ANOVA as appropriate and Tukey (for comparisons with other column means within a group or other column means without grouping) or Šidák (for pairwise comparisons at a given stimulus concentration) corrections for multiple comparisons, as indicated using GraphPad Prism 6.07 (San Diego, CA, USA). Statistical differences were considered significant at *p* < 0.05.

## 3. Results

### 3.1. Siglec-6 Is Consistently and Selectively Expressed on Mast Cells

While the functional consequences of engaging CD33 (Siglec-3) and Siglec-8 on MCs have been explored in depth, the function of Siglec-6 on MCs has not been thoroughly studied in the 15 years since it was found to be highly expressed on CD34+ cell-derived human MCs [7]. In order to examine the biology of this receptor, we first sought to assess surface expression levels of Siglec-6 across a variety of MC types. We found that Siglec-6 was highly expressed (approximately 100,000 receptors per cell) on primary human skin-derived MCs (HSMCs), and that levels remained fairly stable in culture as MC numbers and purity increased up to at least 10 weeks of culture (Figure 1A,B). Similar to HSMCs (Figure 1B,C), the MC lines HMC-1.2, LUVA, ROSA KIT^WT^, and ROSA KIT^D816V^ (Figure 1D–G) all consistently expressed high levels of Siglec-6, even in the presence of gain-of-function mutations associated with MC malignancies. In contrast, the MC lines ROSA KIT^WT^ and ROSA KIT^D816V^ lacked cell-surface expression of Siglec-8 (Figure 1F,G). As previously described, the acute myeloid leukemia cell lines THP-1 (Appendix A) and U937 (Appendix A) also expressed cell-surface Siglec-6 at high levels [16].

Siglec-6 expression has been reported, albeit inconsistently, on non-malignant leukocytes other than MCs, including circulating B cells [16] and plasmacytoid dendritic cells [39], although a recent proteomic study has challenged that finding by suggesting that, besides placental trophoblast expression, Siglec-6 is MC-specific [19]. Employing single-cell RNAseq on biopsy samples from two distinct cohorts of patients with eosinophilic esophagitis (EoE), in which MCs are overrepresented in the tissue, *SIGLEC6* transcript was only detected in MCs (Figure 2). The *SIGLEC8* transcript is likewise limited to the MC population due to the fact that eosinophil mRNA is lost during analysis by scRNAseq. Notably, this is true in control biopsy samples from patients without EoE as well as those from patients with active or inactive EoE (Figure 2). Taken together, these data establish that Siglec-6 is a highly specific marker of human MCs in this tissue.

### 3.2. Siglec-6 Is an Endocytic Receptor with Delayed Kinetics Compared to Siglec-8

Receptor stability at the cell surface and the ability to internalize therapeutic cargo are important factors that determine optimal targeting strategies, as has been reported for Siglec-8 [38]. To date, nothing is known regarding the endocytic properties of Siglec-6. Using a delayed secondary staining method as previously described [38], we find that Siglec-6 is indeed internalized by antibody engagement in HSMCs as well as HMC-1.2 and ROSA KIT^D816V^ cells (Figure 3A,B). We furthermore find no evidence of receptor shedding following ligation using fluorophore-conjugated antibody (Figure 3A). Antibody-engaged Siglec-6 remains at the cell surface longer—and is less completely internalized—than similarly bound Siglec-8 on the same HSMCs; i.e., Siglec-6 endocytosis proceeds more slowly and is less extensive (Figure 3B,C), suggesting that its inhibitory effects on MCs may be more prolonged.

### 3.3. Siglec-6 Engagement Inhibits Both ITAM-Bearing and G-Protein Coupled Receptor-Mediated Activation

To determine whether antibody engagement of Siglec-6 reduces the activation of HSMCs, we incubated HSMCs with anti-Siglec-6 mAb for 30 min prior to stimulating the cells through FcεRI, the C5a receptor, or MRGPRX2. At lower concentrations of a stimulating antibody to FcεRIα, anti-Siglec-6 treatment reduced the amount of MC degranulation as demonstrated by decreased β-hexosaminidase release (Figure 4A and Appendix A). Siglec-6 antibody engagement was similarly effective at reducing MC degranulation in response to C5a (Figure 4B and Appendix A) and compound 48/80 (Figure 4A and Appendix A). To determine if the inhibitory effect depended on the relative timing of Siglec-6 engagement and activating stimulus, we pre-incubated the HSMCs with anti-Siglec-6 for up to 4.5 h prior to stimulation with anti-FcεRIα. The timing of Siglec-6 ligation prior to stimulation had no impact on its inhibitory effects up to 4.5 h, indicative of a prolonged inhibitory effect (Figure 4D). These data suggest that engagement of Siglec-6 causes sustained inhibition of MC secretion responses triggered by both ITAM-dependent and ITAM-independent pathways.

### 3.4. Siglec-6 Co-Engagement with Activating Receptors Enhances Inhibition

To determine whether co-engaging Siglec-6 with an activating receptor could enhance its inhibitory function, as has been shown for both CD33 and Siglec-8 on MCs [10,12], we employed a strategy to co-aggregate Siglec-6 and FcεRIα on peripheral blood CD34+ cell-derived MCs using a secondary cross-linking antibody. This demonstrated that Siglec-6 co-engagement effectively reduced MC activation as measured by CD63 and LAMP-1 expression across a range of concentrations of the stimulating antibody (Figure 5A,B). Co-clustering of Siglec-6 with FcεRIα reduced the peak proportion of CD63+ and LAMP-1+ MCs, and also shifted the response curve such that approximately four-fold more stimulating antibody was needed to activate a similar proportion of MCs (Figure 5A,B). Using a single concentration of 150 ng/mL of anti-FcεRIα antibody, co-aggregation of the receptor with Siglec-6 reduced CD63 upregulation, tryptase release, and secretion of MIP-1β, IL-8, and IL-1β (Figure 5C–G and Appendix A) relative to an isotype control mAb. To investigate whether Siglec-6 co-engagement dampens activatory signaling, we assessed the phosphorylation of ERK1/2 and p38 downstream of FcεRI following co-engagement. Phospho-ERK1/2 levels were diminished in a similar manner to the reduction of CD63 upregulation (Figure 5H,I), and phospho-p38 levels were reduced to baseline by co-engagement with Siglec-6 (Figure 5J), findings consistent with inhibition of kinase activity due to the recruitment of protein tyrosine phosphatases.

To explore whether Siglec-6 possesses inhibitory activity on other cell types with other ITAM-bearing receptors, we co-incubated THP-1 cells with anti-FcγRII and anti-Siglec-6 in the presence or absence of a secondary cross-linking antibody. Binding of primary antibody to FcγRII alone did not induce activating signaling as measured by changes in intracellular phosphotyrosine levels, but aggregation of the receptor with secondary antibody treatment induced measurable protein tyrosine phosphorylation (4), consistent with previous studies [40,41]. Co-aggregation of Siglec-6 and FcγRII significantly reduced phosphotyrosine levels compared to those induced by FcγRII aggregation alone (Appendix A), consistent with ITIM-based inhibitory consequences mediated via recruited protein tyrosine phosphatases.

To account for the possibility that anti-Siglec-6 mAb reduces MC activation by altering the extent of FcεRI-activating cross-linking rather than via inhibition of cytoplasmic signaling events, we produced streptavidin (SAv) tetramers with various fixed ratios of anti-Siglec-6 to anti-FcεRIα (or their respective isotype control mAbs) and incubated HSMCs with these reagents. Siglec-6 aggregation with FcεRIα reduced MC activation as measured by LAMP-1 upregulation (Figure 6A–C), particularly at higher ratios of anti-Siglec-6 to anti-FcεRIα (e.g., 3:1). As a comparison, free antibodies at the same concentrations as those in the 1:1 SAv tetramer treatments were used, although Siglec-6 engagement at this concentration of anti-FcεRIα did not effectively reduce MC activation (Figure 6A–C). Furthermore, detection of unbound FcεRIα epitopes revealed no statistically significant differences in FcεRIα binding in the presence of anti-Siglec-6 or isotype control mAb, indicating that overall antibody binding is not determinative (Figure 6D). Binding of antibody to Siglec-6 is likewise unaffected by the presence of anti-FcεRIα or isotype control mAb (Figure 6E). Stimulated MCs were less capable of responding to repeated stimulation through FcεRIα 24 h after the initial stimulation (Figure 6F, Iso/Anti-FcεRIα). Because MCs treated with SAv tetramers co-engaging Siglec-6 and FcεRIα were prevented from fully responding to the initial stimulation, we tested whether they would respond to secondary stimulation. The results demonstrate that these MCs were similarly unable to respond to secondary stimulation (Figure 6F, Anti-Sig6/Anti-FcεRIα), suggesting that co-engaging Siglec-6 and FcεRIα desensitizes MCs to stimulation through this receptor in the absence of a complete initial response.

## 4. Discussion

Despite the high and selective expression of Siglec-6 on human MCs [7,19], detailed information regarding the function of Siglec-6 on these cells is scarce. In the only publication to our knowledge that demonstrates inhibitory Siglec-6 activity on MCs, Yu et al. show that antibody ligation of Siglec-6 in parallel with FcεRI engagement slightly impedes degranulation of and GM-CSF secretion from CD34+ cell-derived MCs [25]. We extend these findings by demonstrating that Siglec-6 antibody engagement in parallel with exposure to an activating stimulus is capable of inhibiting activation not only through FcεRI but also through C5aR and MRGPRX2. Furthermore, we show that co-crosslinking Siglec-6 with FcεRI using two different approaches potentiates the inhibitory effect of Siglec-6 on MC activation. In addition, the use of primary human MCs from two distinct sources and multiple MC lines ensure that the Siglec-6 properties delineated in this study are not unique to a single system but are instead more broadly shared among human MCs.

The observed inhibition of MC activation through the G protein–coupled receptors C5aR and MRGPRX2 by Siglec-6 engagement indicates broader inhibitory activity than previously appreciated. Several Siglec receptors have been shown to recruit the protein tyrosine phosphatases (PTPs) SHP1 and SHP2 to the cell membrane and halt cellular activation through the activities of these PTPs [10,42,43,44,45], consistent with typical ITIM-based inhibitory signaling. Although Siglec-6 signaling in MCs had not previously been studied, results of Siglec-6 signaling studies in trophoblast cell lines are consistent with PTP-dependent inhibitory signaling. Pervanadate treatment of trophoblast Siglec-6 transfectants leads to the phosphorylation of both the ITIM and ITSM of Siglec-6, and phosphorylated Siglec-6 is able to recruit SHP-2 in these cells [28]. Furthermore, antibody blockade of Siglec-6 on trophoblast cell lines ablates the inhibitory effect of glycodelin A on ERK1/2 phosphorylation, c-Jun expression, and invasiveness [27]. We likewise show that co-crosslinking of Siglec-6 and FcεRIα inhibits the phosphorylation of ERK1/2 and p38, signaling events that are downstream of FcεRI engagement [46]. Additionally, co-crosslinking of Siglec-6 with FcγRII on THP-1 cells also strongly reduces overall phosphotyrosine levels in the cells compared to aggregation of FcγRII alone, indicating that Siglec-6 utilizes a similar inhibitory mechanism in these cells. We hypothesize that the membrane-proximal ITIM is necessary for the inhibitory activity of Siglec-6, consistent with what has been observed for Siglecs-3 (CD33), -7, -8, and -9 [13,47,48], but additional studies are needed to test this hypothesis. Similarly, more experiments are needed to define the mechanisms by which Siglec-6 engagement inhibits signaling via C5aR and MRGPRX2.

Two distinct and complementary approaches were used to co-crosslink Siglec-6 with FcεRIα: (1) the use of a secondary antibody to bind to both the anti-Siglec-6 and anti-FcεRIα on the cell surface; and (2) the use of streptavidin-based tetramers of anti-Siglec-6 and anti-FcεRIα. The caveat of the secondary antibody crosslinking approach is that anti-Siglec-6 may simply displace anti-FcεRIα in these clusters, thereby reducing FcεRI clustering and activating signaling via a mechanism independent of Siglec-6 signaling. The streptavidin tetramer-based method, however, does not permit such displacement, and the staining of unbound FcεRIα shows no differences in FcεRIα binding whether anti-Siglec-6 or an isotype control mAb is used. Together, these results indicate that clustering of Siglec-6 with FcεRI inhibits MC activation, degranulation, and the secretion of chemokines and cytokines. In addition, the data indicate that clustering of Siglec-6 with an activating receptor is necessary for enhanced inhibition of activating signaling. This is consistent with typical ITIM-based inhibitory signaling in which a nearby ITAM-bearing receptor is responsible for the phosphorylation of the ITIM and the recruited PTPs dephosphorylate the ITAM and proximal signaling molecules.

Indeed, studies have demonstrated that clustering of the related Siglec family members CD33, Siglec-7, or Siglec-8 with FcεRI is likewise necessary for effective blunting of MC activation [10,11,12], although under certain circumstances pre-engagement of Siglec-8 is capable of impeding MC activation and systemic anaphylaxis in the absence of co-crosslinking [13,14]. Unlike these Siglecs, however, Siglec-6 is selectively and consistently present on the surface of MCs regardless of tissue source, malignant status, or the presence of the *KIT*^D816V^ mutation. CD33 expression is not selective; it is expressed on myeloid cells beginning at the common myeloid progenitor stage of development but downregulated on mature granulocytes [49,50]. Siglec-7 is also expressed on NK cells, monocytes, dendritic cells, a subset of CD8+ T cells, platelets, and eosinophils [44,51,52,53,54,55] and is largely lost in at least one MC line bearing the *KIT*^D816V^ mutation, HMC-1.2 [11]. Although Siglec-8 is selectively expressed on eosinophils and MCs, we show here that cell-surface Siglec-8 is also lost in two related MC lines, ROSA *KIT*^D816V^ and ROSA *KIT*^WT^. It is not currently understood why Siglec-7 and -8 are lost on some MC lines, while Siglec-6 is consistently expressed across MC subtypes and cell lines [19]. The absence of Siglec-8 on eosinophilic cell lines (HL-60, EoL-3, EoL-1, AML14, and AML14.3D10) and low expression on the MC line HMC-1.1 has previously been noted and attributed to the immature differentiation status of these cells [24,56]. However, the presence of *SIGLEC8* transcript, despite the lack of cell-surface protein in AML14 and AML14.3D10 cells, indicates that a post-transcriptional mechanism such as defective receptor trafficking contributes to the loss of Siglec-8 expression [56].

Plum et al. used proteomics and flow cytometry to demonstrate that Siglec-6 is expressed on all examined human MC populations and they did not detect Siglec-6 on other leukocyte populations [19], in contrast with previous reports of Siglec-6 expression on circulating B cells [16] and plasmacytoid dendritic cells [39]. At the transcriptional level in biopsies obtained from patients with EoE as well as control subjects, there were no B cells or other myeloid cells with a *SIGLEC6* signal above baseline. Therefore, we propose that Siglec-6 is indeed more selectively expressed on human MCs than previously realized, although we cannot rule out expression on leukocyte populations that were not assessed or that may be absent in the tissue biopsies, including acute myeloid leukemia cells [20,21], chronic lymphocytic leukemia cells [22], a subtype of conventional DCs (AS DCs) [57], and tissue-like memory B cells characterized by an exhausted phenotype [58]. Indeed, we confirm Siglec-6 expression on the acute myeloid leukemia cell lines THP-1 and U937 and show that Siglec-6 retains inhibitory activity on THP-1 cells.

Our results demonstrate prolonged desensitization of MCs to FcεRIα stimulation following Siglec-6 co-crosslinking, i.e., MCs fail to respond to anti-FcεRIα 24 h later even in the absence of a complete initial response to stimulation. It is unclear if this is due to the continued presence of anti-Siglec-6/anti-FcεRIα streptavidin tetramers on the cell surface or if these reagents cause the internalization and downregulation of FcεRIα. However, we show that Siglec-6 endocytosis proceeds exceptionally slowly, with the majority of surface-bound antibody remaining after 24 h and a large proportion remaining after 72 h on skin-derived MCs. These data raise the possibility that Siglec-6 may continue to participate in inhibitory signaling long after initial crosslinking and contrast with the more rapid and complete endocytosis of Siglec-8 on the same MCs. Indeed, our results indicate that there is no diminution of inhibition in 4.5 h following pretreatment with anti-Siglec-6, a point at which the majority of antibody-bound receptor remains at the cell surface. It is not yet clear why Siglec-6 remains at the cell surface longer than Siglec-8. It is possible that Siglec-6 is internalized and recycled back to the cell surface without disrupting antibody binding, while Siglec-8 is internalized into the lysosomal compartment and is likely to disengage from the antibody during acidification of the late endosome [38].

Because Siglec-6 co-ligation with an activating receptor is necessary for enhanced inhibitory activity in vitro, we hypothesize that Siglec-6 binds to a glycan or glycans displayed on the ligand of an activating receptor on MCs under physiological conditions. Unfortunately, little is known about Siglec-6 glycan binding preferences or physiological ligands. Upon its initial characterization, Siglec-6 was found to bind to sialyl-Tn but not to Tn antigen, 3′ sialyl-LacNAc, or 6′ sialyl-LacNAc [16]. The glycan binding specificity of Siglec-6 has been refined somewhat by the discovery that Siglec-6, like other Siglecs, requires the carboxyl group of sialic acid for binding but is unique in that it does not require the sialic acid glycerol side chain [26]. It was also discovered to bind to leptin with a reduced affinity relative to the leptin receptor [16]. However, leptin lacks consensus N-glycosylation sequences, and mature leptin is not glycosylated [59,60]; therefore, binding of Siglec-6 to leptin must be sialic acid independent. In contrast, on placental trophoblast cell lines, Siglec-6 binding to glycodelin A is dependent on the presence of sialic acids [27]. The identities of the endogenous Siglec-6 ligands relevant to MC biology and the precise physiological role of Siglec-6 on these cells remain to be determined.

## Figures and Tables

**Figure 1 cells-11-01138-f001:**
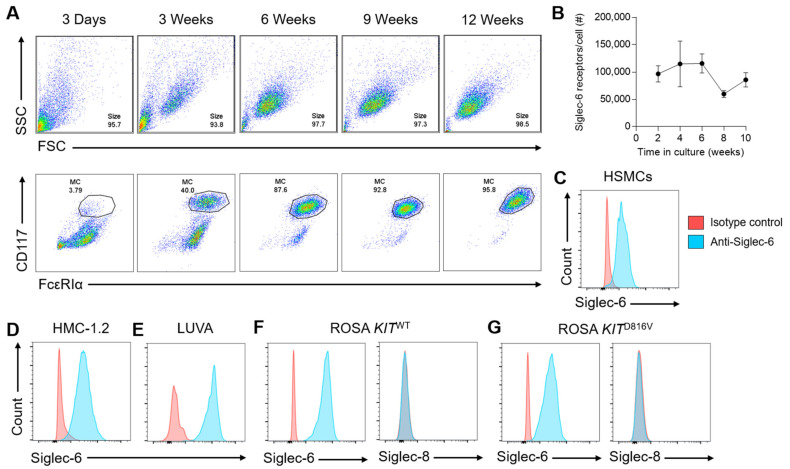
Siglec-6 surface expression on human primary mast cells (MCs) and MC lines. Primary human MCs were isolated and cultured from surgical skin specimens. (**A**) MC purity was determined by flow cytometry based on size, granularity, and expression of the MC markers CD117 and FcεRIα. (**B**) The number of Siglec-6 receptors per MC was determined by quantitative flow cytometry over time in culture. The expression of Siglec-6 on human skin-derived mast cells (HSMCs) (**C**) and the MC lines HMC-1.2 (**D**), LUVA (**E**), ROSA *KIT*^WT^ (**F**), and ROSA *KIT*^D816V^ (**G**) was assessed by flow cytometry relative to staining with an isotype control mAb. Siglec-8 staining, or lack thereof, is also shown for ROSA *KIT*^WT^ (**F**), and ROSA *KIT*^D816V^ (**G**) cells.

**Figure 2 cells-11-01138-f002:**
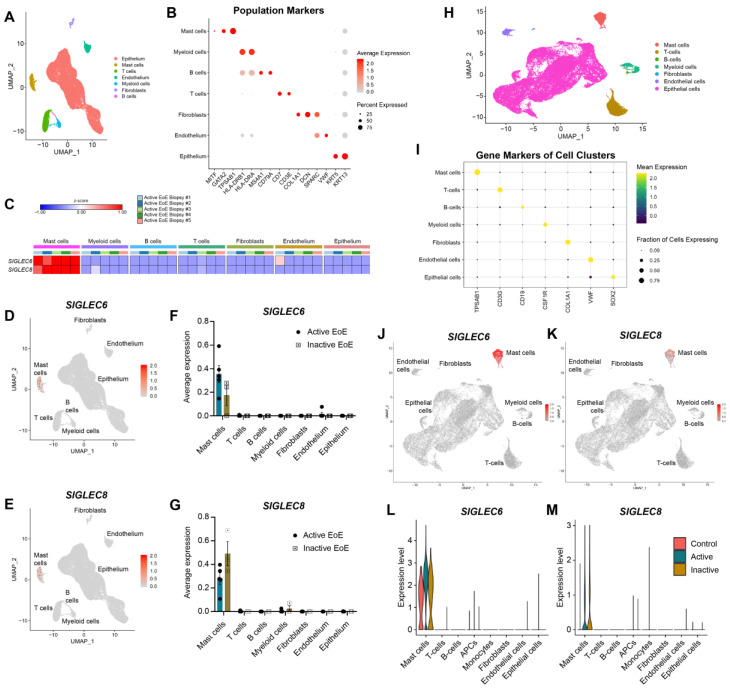
*SIGLEC6* expression is restricted to MCs in esophageal tissue biopsies. Gene expression of esophageal cell populations was assessed by single-cell RNAseq from two distinct cohorts at Cincinnati Children’s Hospital Medical Center (**A**–**G**) and Lurie Children’s Hospital (**H**–**M**). Results were analyzed using an experimental workflow that led to a uniform manifold approximation and projection (UMAP) plot for dimension reduction displaying 39,562 (**A**) or 44,153 (**H**) single cells. The plots are colored by cell types derived from the shared nearest neighbor (SNN) clustering and enriched marker genes. (**B**,**I**) Dot plots of marker genes for indicated cell types are shown. (**C**) Heatmap of *SIGLEC6* and *SIGLEC8* expression level in five donors with active EoE, in which the color for each gene corresponds to the average per-cell gene expression within the given patient in the esophageal cell population. *SIGLEC6* expression in all patients is presented in feature plots (**D**,**J**), average expression by diagnosis is presented in a bar graph (**F**), and expression mean and variance by diagnosis is presented in a violin plot (**L**). *SIGLEC8* expression in all patients is presented in feature plots (**E**,**K**), average expression by diagnosis is presented in a bar graph (**G**), and expression mean and variance by diagnosis is presented in a violin plot (**M**). Cells were isolated from esophageal biopsies derived from patients with active (*n* = 5 (**A**–**G**) or *n* = 7 (**H**–**M**)) or inactive EoE (*n* = 3 (**A**–**G**) or *n* = 4 (**H**–**M**)) or a non-EoE control subject (*n* = 1 (**H**–**M**)).

**Figure 3 cells-11-01138-f003:**
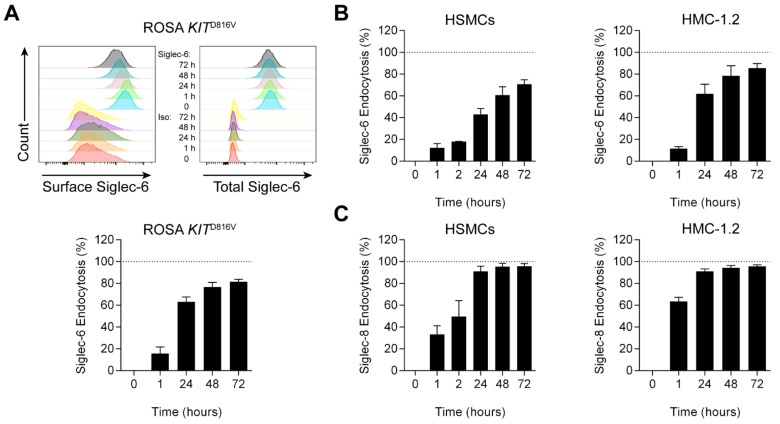
Siglec-6 is stable at the surface of MCs and is slowly internalized following antibody ligation. (**A**) Delayed fluorophore-conjugated secondary mAb detection of anti-Siglec-6 mAb was used to assess Siglec-6 levels remaining on the surface of ROSA *KIT*^D816V^ cells (Surface Siglec-6) following antibody ligation after various durations. Fluorophore-conjugated anti-Siglec-6 mAb was used to assess both cell-surface and internalized Siglec-6 (Total Siglec-6) on ROSA *KIT*^D816V^ cells. Isotype control mAb (Iso) is used as a negative control in both strategies. Because loss of total Siglec-6 was not observed, Siglec-6 endocytosis was calculated based on the loss of cell-surface Siglec-6 on ROSA *KIT*^D816V^ cells (**A**), as well as HSMCs and HMC-1.2 cells (**B**). Siglec-8 endocytosis was similarly determined on HSMCs and HMC-1.2 cells (**C**). Data are representative (histograms) or represent the means and standard deviations of three independent experiments.

**Figure 4 cells-11-01138-f004:**
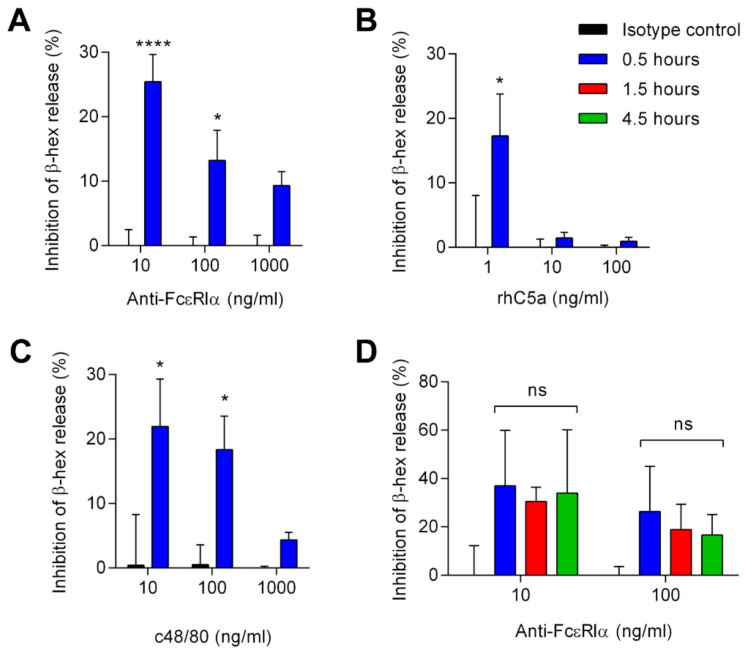
Siglec-6 antibody engagement in parallel with various stimuli inhibits HSMC degranulation. HSMCs were collected, washed, and incubated in the presence or absence of anti-Siglec-6 mAb for the indicated duration of time. HSMCs were then incubated with the indicated concentrations of anti-FcεRI mAb (**A**,**D**), rhC5a (**B**), or compound 48/80 (**C**) for 30 min. Release of β-hexosaminidase was measured by colorimetric assay, and inhibition was calculated on the basis of the reduction of β-hexosaminidase release relative to treatment with an isotype control mAb. Data represent the means and standard deviations of fifteen (**A**), twelve (**B**,**C**), and six (**D**) replicates from five (**A**), four (**B**,**C**), and two (**D**) distinct HSMC cultures. * *p* < 0.05; **** *p* < 0.0001 vs. isotype control at same stimulus concentration; ns, not statistically significant, determined by two-way ANOVA with Šidák correction for multiple comparisons.

**Figure 5 cells-11-01138-f005:**
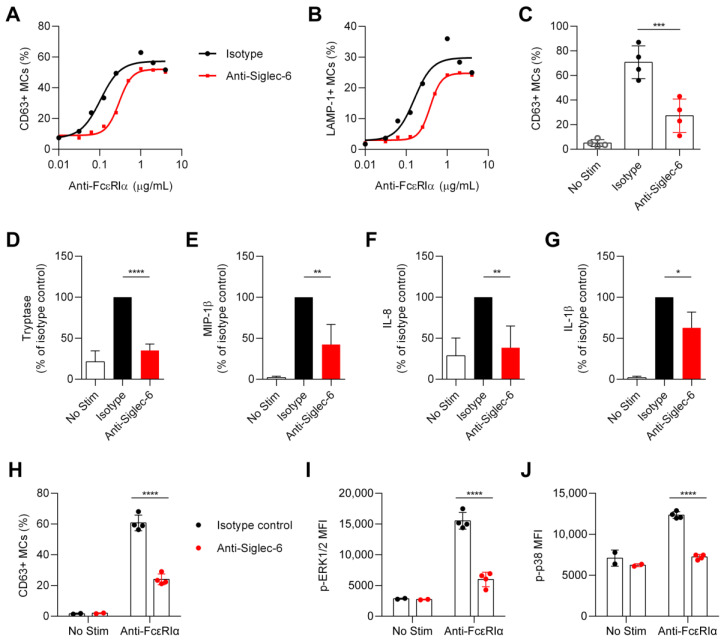
Co-crosslinking of Siglec-6 and FcεRIα on MCs enhances inhibitory activity. (**A**,**B**) CD34+ cell-derived MCs were incubated with the indicated concentration of anti-FcεRIα, as well as anti-Siglec-6 or isotype control mAb. Antibodies were crosslinked using secondary anti-mouse IgG antibody, and cells were incubated at 37 °C for 20 min to permit cell stimulation. The percentage of CD63+ (**A**) or LAMP-1+ (**B**) MCs was then determined by flow cytometry. (**C**–**G**) CD34+ cell-derived MCs were incubated with 150 ng/mL anti-FcεRIα and either anti-Siglec-6 or isotype control mAb, or were not incubated with mAbs (No Stim). Expression of CD63 (**C**) was determined by flow cytometry, and tryptase release (**D**) was measured colorimetrically in cell-free supernatant after 20 min of stimulation. MIP-1β (**E**), IL-8 (**F**), and IL-1β (**G**) were detected in cell-free supernatant after overnight stimulation. Levels were normalized to those measured in the stimulated samples incubated with the isotype control antibody (**D**–**G**). (**H**–**J**) CD34+ cell-derived MCs were incubated in the presence or absence (No Stim) of 150 ng/mL anti-FcεRIα with either anti-Siglec-6 or isotype control mAb. Antibodies were crosslinked using secondary anti-mouse IgG antibody, and cells were incubated at 37 °C for 15 min in complete medium to permit cell stimulation. Following stimulation, cells were fixed and stained for surface CD63 (**H**), intracellular phospho-ERK1/2 (**I**), or intracellular phospho-p38 (**J**). Data are representative (**A**,**B**), represent the means and standard deviations of four (**C**–**E**) or three (**F**,**G**) independent mast cell cultures and experiments, or represent the individual values, means, and standard deviations of two (**H**–**J**: No Stim) or four (**H**–**J**: Anti-FcεRIα) replicates. * *p* < 0.05; ** *p* < 0.01; *** *p* < 0.001; **** *p* < 0.0001 by one-way ANOVA with Tukey test to correct for multiple comparisons (**C**–**G**). **** *p* < 0.0001 by two-way ANOVA with Šidák correction for multiple comparisons (**H**–**J**).

**Figure 6 cells-11-01138-f006:**
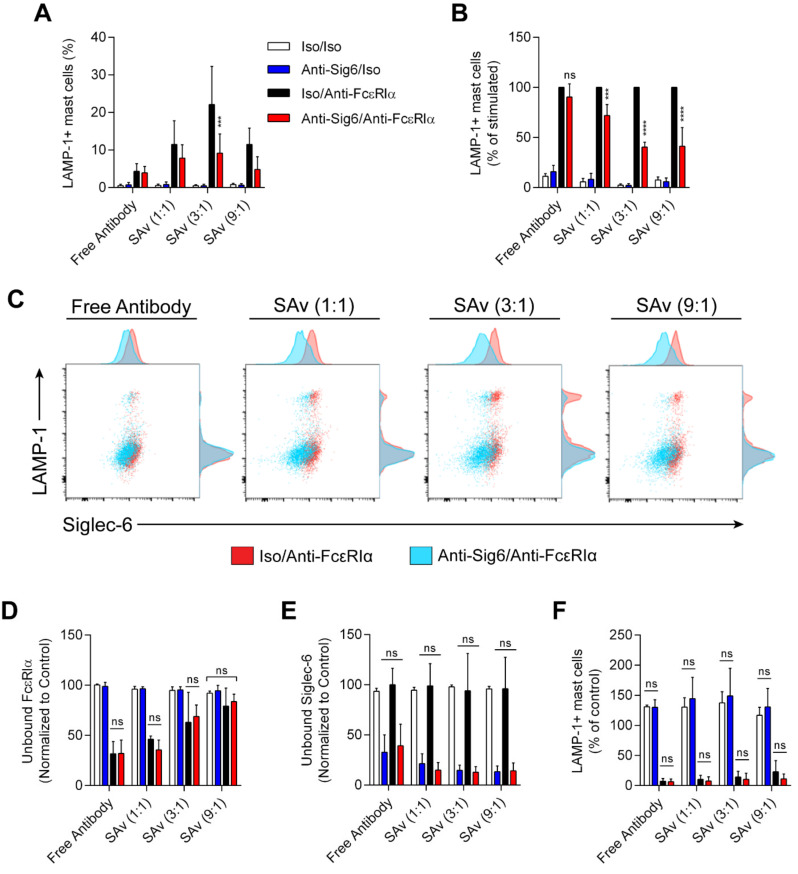
Inhibition induced by co-engagement of Siglec-6 and FcεRIα does not depend on reduced FcεRIα crosslinking. Streptavidin complexes (SAv) with the indicated ratio of anti-Siglec-6 to anti-FcεRIα or uncomplexed antibody with final concentrations matching that of the 1:1 SAv complexes were used to stimulate HSMCs. After 30 min, the cells were washed, stained, and analyzed by flow cytometry. MCs were pre-gated on live CD117+ cells. The percentage of MCs expressing surface LAMP-1 (**A**), LAMP-1 normalized to the Iso/Anti-FcεRIα sample in each group (**B**), and dot plots comparing stimulated samples with anti-Siglec-6 or isotype control mAb with respect to surface LAMP-1 and unbound Siglec-6 expression (**C**) are shown. Unbound FcεRIα (**D**) and unbound Siglec-6 (**E**) normalized to the control are quantified based on sample MFI. (**F**) Approximately 24 h after initial stimulation as indicated, cells were stimulated with 500 ng/mL anti-FcεRIα and stained for LAMP-1 and MC markers. The percentage of LAMP-1+ HSMCs normalized to that of cells not initially treated with antibody is quantified. Data represent the means and standard deviations (**A**,**B**,**D**–**F**) or are representative (**C**) of four independent experiments using three distinct HSMC cultures. *** *p* < 0.001; **** *p* < 0.0001; ns, not statistically significant, determined by two-way ANOVA with Tukey test to correct for multiple comparisons.

## Data Availability

The data presented in this study are openly available in ImmPort under study accession SDY1919.

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
