# Peer review of "Functional and Phenotypic Characterization of Siglec-6 on Human Mast Cells"

_cells, 2022, doi:10.3390/cells11071138_

Round 1

Reviewer 1 Report

This is an important study by Robida PA et al. addressing expression and function of the inhibitory receptor Siglec 6 on human primary mast cells. My major comments are the following: 

1-There was no significant difference in the inhibitory property of Siglec 6 between 30 min or 4.5 hours of “pre-treatment” with Siglec 6 antibody. Is the amount of Siglec 6 endocytosis at 4.5 hours not significant? Would the 24 hour pre-treatment diminish Siglec 6 inhibition because of receptor internalization? 

2- Is it possible that the difference in binding/internalization kinetics is due to differences in the antibody binding properties? Do the Siglec 6 and Siglec 8 antibodies referenced here have similar affinities for their targets or are their epitopes on comparable domains of the receptor?

3- Does the cross linking of Siglec 6 and FCeRI also cause endocytosis of both receptors in the long term? This may also be indicated by the prolonged desensitization phenotype. Perhaps stimulating with a different, FCeRI-independent, mechanism could show whether it is a global effect.

6-Fig. 5D-G: Do the authors see inhibition for trytase and cytokine release by engaging Siglec 6 without cross linking FceRI and Siglec 6? Also, stats are missing.

4- Fig. 6F seems to indicate that the inhibition of secondary stimulation is not dependent on primary Siglec 6 stimulation as group 3 (black bars, iso/anti FceRIa) shows just as much secondary inhibition as group 4 (red bars)

5- Do the authors expect the effects of not cross-linked Siglec 6 on MAPK signaling and cytokine release to be similar to those shown for the cross-linked experiments? 

Minor comments:

 1. Authors show and mention in the legend of Fig. 3 that receptor shedding is not significant. It would also be nice for this result to be explicitly stated in the results section for the figure.

2. Can the authors show the % of beta hex release in addition to % of inhibition in Fig. 4?

3. Although not significant, the drop in expression at 8 weeks of culture seems substantial. Was the data for B gated or taken from all cells? If from all cells, might this decrease be due to higher expression on immature relative to mature mast cells?

4-Line 290: authors mention that data establish that Siglec 6 is a highly specific marker of human mast cells. This seems to be applicable only to mast cells in tissues from EoE. Unclear, whether this can be generalized to other tissues and/or other conditions. 

Author Response

Thank you for your helpful comments. We have revised the manuscript to satisfy your concerns and believe that the manuscript is much improved as a result. Our detailed responses can be found below:

Major:

  1. The association between Siglec internalization and signaling is intriguing. Some data, including our own, suggest that inhibition of internalization can inhibit Siglec-8 signaling, at least in eosinophils. This has been harder to explore in mast cells because drugs used to explore this often have their own direct effects on mast cell secretion responses. While we have not performed experiments to evaluate Siglec-6 internalization at precisely 4.5 hours on skin-derived MCs, the results we have obtained from time points near this time indicate that only a relatively minor percentage (likely close to 20-25%) of the bound Siglec-6 has been internalized by this time. We have not performed experiments involving pretreatment with anti-Siglec-6 as far as 24 h (or longer for that matter) before stimulation, although the endocytosis results indicate that approximately 60% of the bound Siglec-6 would be available to participate in inhibitory signaling at the 24 h time point. The reviewer’s question is quite germane, especially as it relates to how an anti-Siglec-6 antibody might behave in vivo regarding its inhibitory effects. This and related questions are currently under investigation but are beyond the scope of this present work. However, in light of these important issues, we have added statements to the Discussion to clarify this at lines 642-645 (please note that line numbers are given for the manuscript version with changes tracked).
  2. We have not measured the affinities or other binding properties of our anti-Siglec-6 and anti-Siglec-8 antibodies. So antibody binding properties cannot be definitively ruled out with regard to effects on Siglec internalization kinetics or inhibitory activity; however, our previous study indicated that Siglec-8 was internalized with similar kinetics and to a similar extent using two antibody clones that bound to distinct epitopes on different extracellular domains of the receptor (O’Sullivan et al., 2018). Due to the limited commercial availability of antibodies selective against Siglec-6 that show no binding to other Siglec family members, we have not been able to repeat the aforementioned experiment with other antibodies against Siglec-6.

  3. Because of the manner in which Siglec-6 and FcεRI were crosslinked (with blockade of detecting antibody epitopes), it is difficult to assess FcεRIα expression or endocytosis under these circumstances. However, because FcεRI has been reported to be relatively rapidly internalized in response to receptor crosslinking, we would expect that, in the context of stimulation, receptor co-aggregation with Siglec-6 would not further promote FcεRI endocytosis and may even prevent its rapid internalization through its inhibitory activity. This may prevent FcεRI recycling to the cell-surface and prolong desensitization to additional stimulation as you suggest, although this will require further study. Although the experimental duration of 30 min may be insufficient for this purpose, we show in Fig. 6D-E that aggregation of Siglec-6 by itself or with FcεRIα does not significantly alter unbound FcεRIα levels on the surface of the MC and that unbound Siglec-6 levels are not significantly altered by FcεRIα crosslinking/co-crosslinking.

  4. We apologize for the confusing explanation in the manuscript. We intended for Fig. 6F to show that mast cells are incapable of responding to a secondary stimulation through FcεRIα in 24 hrs (iso/anti-FcεRIα) and that impeding the initial activation (anti-Siglec-6/anti-FcεRIα) does not alter the response to the secondary stimulation, i.e., there was neither a complete response during the primary stimulation nor a response during the secondary stimulation. The wording for this explanation has been clarified on lines 525-531 of the Results and 627-629 of the Discussion.   

  5. We expect that pre-treatment with anti-Siglec-6 in the absence of crosslinking with FcεRIα would reduce signaling events downstream of FcεRI to an extent roughly proportional to its inhibition of mast cell activation. Therefore, we would expect the inhibition of signaling to similarly be much reduced relative to co-clustering Siglec-6 and FcεRI.

  6. We have not examined the inhibition of tryptase or cytokine release from pre-treatment with anti-Siglec-6 before MC stimulation but expect that the tryptase data would mirror the beta-hexosaminidase release data in Fig. 4. Because the responsiveness and peak levels of these proteins differ somewhat between mast cell cultures from distinct donors, we have normalized released protein levels for each culture prior to statistical analysis. These normalized figures are now found in the revised Fig. 5 (panels D-G), with statistical notation, and the panels demonstrating non-normalized protein concentrations have been moved to Fig. S3 in the Supplemental Information.

Minor:

  1. We agree that this result should be mentioned in the Results section. An explanation of this result has been included in the revised manuscript on lines 387-388.
  2.  We have included % beta-hex release for each stimulus and each concentration in the new Fig. S2.
  3. The data from panel B is based on Siglec-6 expression on cells in the mast cell gate (CD117+ FcεRIα+ cells). This reduction, which does not appear to be sustained, may be related to a particularly vigorous proliferative burst of the MCs in culture just prior to the analysis. Based on our observations that Siglec-6 is highly and consistently expressed on MCs both before and after this particular time point, we have attributed this perturbation to culture conditions and/or statistical noise rather than to differentially regulated expression of Siglec-6 at distinct stages of MC development.

  4. The data indeed assess Siglec-6 expression on cells isolated only from esophageal biopsies (from both healthy control subjects and active/inactive EoE patients) but are consistent with other published results such as Plum et al. Nevertheless, the statement on line 380 of the revised manuscript has been changed to clarify this.

Reviewer 2 Report

`In their study, Robida and colleagues examined the inhibitory effects induced by Siglec-6 engagement in human mast cells. An interesting and well-performed investigation.

However, some concerns remain:

  1. Two human mast cell differentiation systems, namely from skin and blood progenitors, are used in this study. It is unclear whether the two are different when it comes to Siglec-6-mediated inhibition of degranulation. The authors are invited to discuss this matter and provide exemplary evidence of differences and commonalities.
  2. Fig. 4 illustrates the effectiveness of anti-Siglec-6 antibodies in inhibiting HSMC degranulation induced by various stimuli. To understand the level and extent of anti-Siglec-6 antibody-mediated inhibition, the authors should show the percentage of degranulation induced in HSMCs by anti-FceRIa, C5a and C48/80 treatment.
  3. In their paper, Yanase and colleagues (2021) demonstrate that the concentration required for inducing HSMC degranulation by C5a is 10ng/ml. Differently from what the authors state in their manuscript, these findings would suggest that Siglec 6-engagement does not affect C5a-induced mast cell degranulation. At which concentration do the authors observe a significant C5a-induced HSMC degranulation?
  4. Furthermore, the authors should provide evidence about the lasting effect of the inhibition.
  5. It seems that in the authors' hand, LAMP1 is a less suitable marker to detect mast cell degranulation than CD63. Why do the authors use LAMP1 for the experiments displayed in Fig. 6?
  6. The % of mast cell degranulation induced by anti-FceRIa (150ng/ml) reported in Fig 6 panel A and C are very different. How do the authors explain this matter?
  7. Finally, the authors should indicate the number of mast cell cultures used (in the figure legends).

Author Response

Thank you for your kind assessment and thoughtful insight. We have revised the manuscript to address your concerns and believe that the manuscript is much improved as a result.

  1. Thank you for noting this. Indeed, we show similarly robust inhibition of FcεRI-mediated mast cell activation using both skin-derived and blood progenitor-derived MCs. We also do not observe any apparent differences in Siglec-6 expression based on staining MFI, and internalization of Siglec-6 appears to be consistent between not only skin- and blood progenitor-derived MCs but also the MC lines HMC-1.2 and ROSA KITD816V. We propose that Siglec-6 is expressed and functions similarly on both skin- and blood progenitor-derived MCs and that the inclusion of MCs from both sources makes the findings of the study stronger and more universal. We have added a statement to this effect in the Discussion on lines 543-546 (please note that line numbers are given for the manuscript version with changes tracked).
  2. Bar graphs presenting the net beta-hexosaminidase release for each stimulus and concentration have now been included as Fig. S2 in the Supplemental Information.

  3. Thank you for bringing this to our attention. Our results closely resemble those described by Yanase et al., but the beta-hexosaminidase release at 1 ng/ml C5a is in fact statistically significant in our experiments. However, similar to the published findings, it is modest. This is shown in the new Fig. S2 in the revised Supplemental Information.

  4. We have demonstrated that pretreatment with anti-Siglec-6 as far as 4.5 h prior to stimulation through FcεRI exerts the same inhibitory effect as pretreatment 0.5 h prior to stimulation (Fig. 4D). We additionally show that MCs treated with streptavidin tetramers co-engaging Siglec-6 and FcεRIα fail to respond fully to the initial stimulation and are similarly desensitized to a secondary stimulation at 24 h as those that responded fully to the initial stimulation (Fig. 6F). It is unclear yet whether Siglec-6 continues to participate in inhibitory signaling at this time point, but the endocytosis data we include in Fig. 3B suggest that the majority of the antibody-bound Siglec-6 remains at the cell surface. Additional discussion regarding whether endocytosis may impact Siglec-6 function can be found in our response to the first question posed by Reviewer #1.

  5. We apologize for the inconsistent use of activation markers. We have observed that Siglec-6 ligation or co-ligation reduces mast cell activation as assessed by beta-hexosaminidase release, surface CD63 expression, surface LAMP-1 expression, tryptase release, and the secretion of MIP-1β, IL-8, or IL-1β. We were unable to establish a single marker of mast cell activation that was more sensitive to inhibition by Siglec-6 or more relevant. As we indicate in Fig. 5A,B, both CD63 and LAMP-1 surface expression respond well to Siglec-6 inhibition.

  6. The anti-FcεRIα response curves differ somewhat between CD34+ cell-derived MC cultures from different donors, although the concentration of 150 ng/ml was consistently in the linear portion of the curve. Because panel A represents a single MC culture and panel C includes data from 4 additional MC cultures, the data do not align perfectly.

  7. Thank you for the comment. This information has now been included in the figure legends for Figs. 5 and 6 as well as Figs. S2 and S3.

Round 2

Reviewer 1 Report

No additional comments.